# BAYESIAN TREE-DEPENDENT FACTORIZATION

## ABSTRACT

We propose Bayesian Tree-dependent Factorization (BTF), a novel probabilistic representation learning model that uncovers hierarchical, continuous latent factors in complex datasets. BTF constructs a tree-based model that discovers interpretable factorizations of the data wherein each factor has a conditional relationship to its parent, allowing it to capture both global and local effects. This approach is particularly well-suited for biological data, where traditional methods like PCA fail to capture higher-order dependencies and hierarchical structure. A significant contribution of this work is the multi-view extension of BTF, which allows for the joint analysis of multiple data modalities. By learning shared loadings across views while maintaining distinct factors for each modality, multi-view BTF improves performance and enables deeper insights into the relationships between different data types. We demonstrate the performance of BTF in simulations as well as in a real-world application to gene expression and clinical data in breast cancer patients, revealing biologically and clinically meaningful patient trends, and showing that BTF is a valuable representation learning tool for analysis and hypothesis generation.

## 1 INTRODUCTION

Complex data are often comprised of both global effects that apply broadly across a significant portion of the data as well as local effects that may be detectable only in small subsets. This is especially the case in biological datasets; for example, single cell RNA sequencing counts are widely influenced by effects such as cell size or cell cycle phase, but certain subsets of samples may exhibit specific properties related to more targeted effects such as cell-type-specific stress responses. Similarly, breast cancer patients are all subject to certain common biological effects such as tumor growth, but certain subtypes of patients may be particularly affected by biological responses related to specific biological or clinical markers such as immunohistological status. There may be even further effects conditioned on interactions between immunohistological status and other processes such as metastasis, stage or particular immune response. A significant challenge presented by these types of data is that while it can be useful to discretize certain categories or sub-groups of samples, many of the significant latent effects are inherently continuous in nature. These considerations motivate a highly interpretable approach that learns a dependent structure of continuous factors so that we are able to jointly infer and understand the continuous effects present in the data and their relationships to each other.

Common factor analysis approaches such as Principal Component Analysis and Independent Component Analysis often do not provide interpretable factors, even when sparse; every principal component typically captures many different sources of variance to different degrees, without accounting for higher order dependencies between effects. These approaches also fail to account for any kind of hierarchical structure and instead impose strong orthogonality or independence constraints. More recent methods attempt to address these weaknesses by directly inferring hierarchical structure under certain strong constraints. One such approach, Tree-Dependent Component Analysis (Bach & Jordan, 2012), relaxes the orthogonality assumptions made by more classic factorization methods by inferring a set of components that are well fit by a tree-structured graphical model. Even more recently, hierarchical approaches to matrix factorization have come into increased focus. These also do not impose orthogonality constraints, but they often make discrete assumptions of hierarchical structure membership that do not allow for fully continuous effects. (Sugahara & Okamoto, 2024; Almutairi et al., 2021; Li et al., 2019) One relevant example of this approach is eTrees (Almutairi

et al., 2021), which generates tree-structured embeddings that explicitly encode a hierarchical structure by assuming discrete binary indicators of category and subcategory-membership. Other common approaches to the unsupervised discovery of hierarchical structure in data often rely on two broad techniques: recursively constructed sampling-based approaches such as hierarchical Dirichlet processes (Teh et al., 2004); or post-hoc modeling of the hierarchical structure using agglomerative (or similar) clustering of the results of factorization methods with desirable properties. Most of these approaches also assume that the underlying hierarchy leads to discrete groupings and do not capture the kinds of continuous effects we previously described.

We propose a Bayesian approach to a unified factor analysis of hierarchically structured yet continuous effects. In contrast to all the previously described methodologies, Bayesian Tree-Dependent Factorization infers a dependent structure that models continuous effects and also generates an explicit interpretation of each component's effect given their parents. Notably, one common machine learning approach that incorporates non-linear dependencies in a hierarchical fashion is the decision tree (more specifically, classification and regression trees)(Breiman et al., 1984). Decision trees can be used for both supervised and unsupervised learning tasks, but usually rely on discrete splits and are prone to overfitting. To address these problems, we typically rely on ensembles of decision trees (or random forests). (Breiman, 2001) While these ensembles also leverage hierarchical structure in order to make predictions and are very robust, they sacrifice the interpretability of the decision tree. This challenge is one of the motivations of the tree-based approach to factorization that we explore here.

A key contribution of this work is its emphasis on flexibility and interpretability. Because we have taken a Bayesian approach, we can place priors on the factor loadings as well as the factors themselves, and generate posterior estimates of uncertainty and likelihood. We also place certain constraints on the factor loadings which allow us to more easily interpret factor dependencies as conditional weights.

Finally, we introduce a multi-view extension to BTF. Multi-view datasets are becoming increasingly prevalent, particularly in genomics and healthcare applications, where the integration of different dependent modalities can be crucial to understanding the complex underlying mechanisms. By leveraging the structure present in multiple modalities of data, we can increase the robustness of our estimates across the views. The relationships between the modalities also give us the opportunity to interpret what we learn in one view in the context of another. We demonstrate that this interpretability can significantly improve our understanding of the latent structure in clinical and biological data, and that we can leverage it to observe detailed mechanistic relationships between the signals in each view and formulate powerful hypotheses.

## 2 METHODS

BTF is a Bayesian approach which extends the probabilistic PCA (Tipping & Bishop, 1999) formulation with a hierarchical loading structure. As in probabilistic PCA, samples are drawn from Gaussian distributions with means generated from a linear combination of independently drawn Gaussian factors and a shared tolerance or noise parameter $\sigma$. As such, the probabilistic graphical model for BTF (shown in Fig. 1) closely resembles that of probabilistic PCA. In contrast to probabilistic PCA, we formulate the loadings on the factors in BTF such that they are dependent on one another. Each loading is the product of sub-loadings $z$, some of which they share with other loadings. A binary tree determines the sub-loadings used to construct a given factor's loading and the way in which they are combined, as depicted in Fig. 2. Crucially, the position of a node sub-loading in relation to its parent dictates whether it is multiplied by the parent's sub-loading $z_p$ or the complement $(1 - z_p)$, indicating that right and left child factors describe conditional effects whose distributions are positively and negatively correlated with the effect distributions of their parent, respectively. As a result, we hypothesize that the BTF model is well-suited to data where hierarchical dependencies involving up- or down-regulation are of particular relevance (as is often the case in biological data).

The depth of the binary tree is specified *a priori* and may be specific to the chosen application. We further constrain the values of the sub-loadings such that they fall between 0 and 1, which lends itself to a simple interpretation of a sample's factor sub-loading $z$ as the conditional weight of that factor in that sample given the weight of its parent factor.

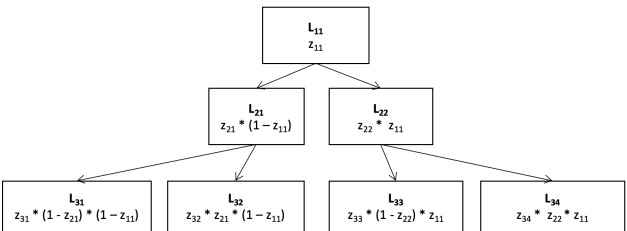

Figure 1: A visualization of the tree structure used to form the loadings for Bayesian Tree-dependent Factorization out of the sub-loadings. The binary tree demonstrates the dependencies induced by the loading composition.

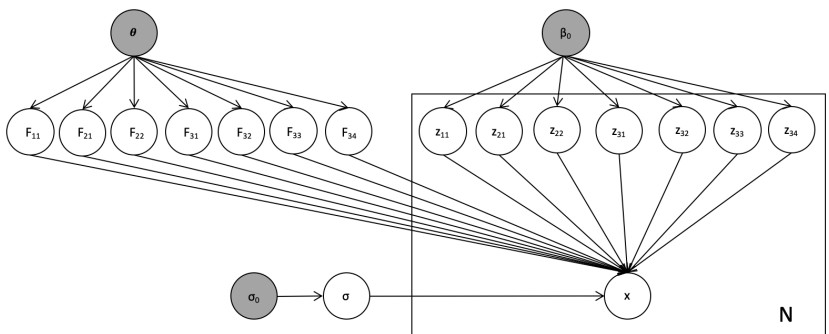

Figure 2: A plate-notated visualization of Bayesian Tree-dependent Factorization (BTF) with a depth of 3. The tree may be of arbitrary depth depending on user specification. Gray nodes denote variables known *a priori*.

## 2.1 MODEL

We define the following latent variables:

- Let $\theta$, $\beta$ and $\sigma_0$ be hyperparameters.

- Let $x_i$ with $i \in 1...N$ be the data with $D$ dimensions.

- Let $F_{ij}$ be the $D$-dimensional factor $j$ at depth $i$ in the binary tree.

- Let $z_{ij}$ be the $N$-dimensional sub-loading $j$ at depth $i$ in the binary tree.

- Let $\sigma$ be the global noise parameter.

The prior distributions on the latent variables defined by BTF can be summarized as follows:

$$F_{ij} \sim \mathcal{N}(0, \theta)$$
$$z_{ij} \sim \beta(\beta_0)$$
$$\sigma \sim \mathcal{N}(0, \sigma_0)$$

Note that the number of factors and loadings defined by the model depends on the depth of the tree. Should a depth of 3 be specified, data would be sampled from the following conditional probability distribution:

$$p(X|Z, F) \sim \mathcal{N}(\mu_x, \Sigma_x)$$
$$\mu_x = z_{11} * F_{11}$$
$$+ z_{21} * (1 - z_{11}) * F_{21}$$
$$+ z_{22} * z_{11} * F_{22}$$
$$+ z_{31} * (1 - z_{11}) * (1 - z_{21}) * F_{31}$$
$$+ z_{32} * (1 - z_{11}) * z_{21} * F_{32}$$
$$+ z_{33} * z_{11} * (1 - z_{22}) * F_{33}$$
$$+ z_{34} * z_{11} * z_{22} * F_{34}$$

where we have visually indented the contributions of each level of the tree to the overall sample means in order to demonstrate the relationship between the depth parameter and the terms that make up the conditional likelihood. We employ Adaptive Moment Estimation (ADAM) (Diederik, 2014) for the learning process.

## 2.2 MULTI-VIEW BAYESIAN TREE-DEPENDENT FACTORIZATION

To increase the stability and interpretability of BTF, we also formulate a multi-view extension to the model. One of the possible challenges to factorization problems and to BTF in particular is identifiability; in particular, the large number of sub-loadings specified in the BTF model means that there are many more degrees of freedom in the parameter space than there are factors. Due to the Bayesian nature of the model, stronger or additional priors may be added to alleviate these concerns.

In particular, we integrate multiple views of data by jointly optimizing the sum of the log likelihoods of multiple factorizations across a common set of samples with different data types. We constrain the loadings across views such that they are equal. As such, we learn one set of loadings, but as many sets of factors as there are views. This significantly constrains the loading parameters and ensures that the factors across views are related, which further assists in our interpretation. Additionally, we incorporate weight parameters on each of the individual model log likelihoods which may be specified *a priori* in order to up-weight the effect of any given view.

## 3 SIMULATIONS

In order to test the ability of the BTF to recover a ground truth hierarchical structure, we first applied BTF to data from two different simulations with three levels of effects. We devise simulations by randomly sampling 7 sets of sub-loadings from a uniform distribution between 0 and 1 and then composing the loadings from the sub-loadings in the hierarchical fashion specified by the BTF model. To induce hierarchical structure in the data, we mask each set of sub-loadings such that the right-child sub-loadings of any given parent loading are only non-zero when the parent loading is greater than 0.5, and left-child sub-loadings are non-zero only when their parent loading is less than 0.5. We sample $N$ sub-loadings at each level and combined these with arbitrarily chosen factors that demonstrate spread along $D$ dimensions to produce a simulation of $N$ samples in $D$ dimensions. We demonstrate this generative process in an easy-to-visualize simulation in two dimensions (Fig. 3). All experiments with simulated data are run on simulations of 1,000 samples.

We ran BTF on the simulated data and then compared the inferred loadings to those recovered by three baseline methods: Principal Component Analysis (PCA), Independent Component Analysis (ICA) (Lee & Lee, 1998), and Tree-dependent Component Analysis (TCA). PCA and ICA are the most common approaches to factorization and make basic assumptions regarding orthogonality or independence of the components, respectively. TCA relaxes these assumptions by allowing for a set of components that are well fit by a tree-structured graphical model. As such, we consider it a comparable approach for the purposes of inferring an unknown hierarchical structure.

One of the advantages of BTF is that it is able to learn an arbitrary number of factors without being constrained by the rank of the data. PCA and the commonly used FastICA (Hyvarinen, 1999) algorithm are each only able to generate a maximum of $r$ factors where $r$ is the rank of the given

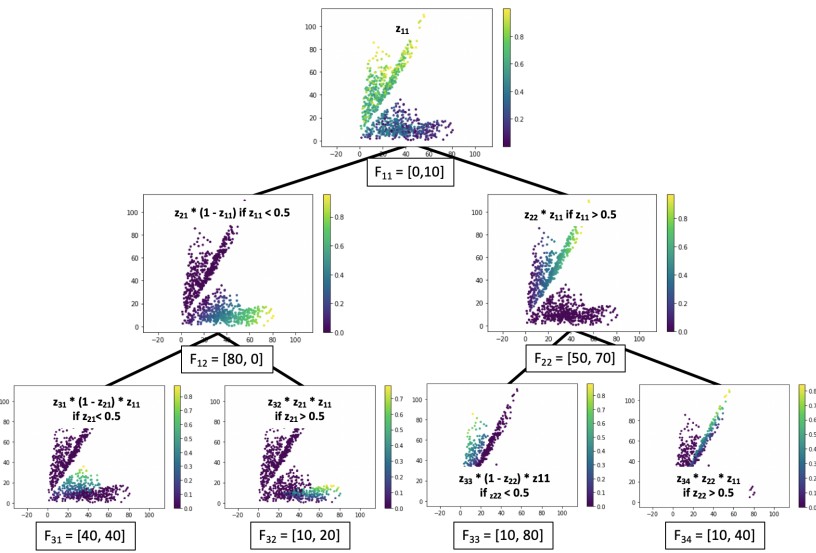

Figure 3: A visualization of the effects generated in the 2-dimensional simulation, arranged in the hierarchy induced by the generative loading structure.

data. In order to ensure a fair comparison, we also ran the models on a simulation in 7 dimensions which closely resembles the hierarchical structure we used for the 2 dimensional simulation, but with the effects of each level of the hierarchy represented in new dimensions plus additional small noise added in extra dimensions to ensure that the data is full rank.

## 3.1 EVALUATION OF SIMULATIONS

In order to numerically evaluate the quality of the factorizations learned by BTF in contrast to the chosen baseline methods, we examined the Spearman correlations of each of the true factor loadings used in the generative simulations to each of the factor loadings learned by each method. We also visualized the 2-dimensional data colored by the learned loadings and compared these to the ground truth visualization in order to evaluate the quality of the effects captured by BTF.

## 3.2 RESULTS OF SIMULATION ANALYSIS

Fig. 4 shows the correlations between the learned loadings and the ground truth loadings. The heat map shows that the correlations are very high along the diagonal, demonstrating that BTF is able to recover the correct structure. Additionally, the high correlation block structure away from the diagonals demonstrates that all recovered loadings are highly correlated with their ancestor loadings in the loading tree. This is consistent with our expectation regarding the structure of the sub-loading compositions and further validates that BTF is able to learn interpretable dependencies between the loadings. Fig. 4 shows a visualization of the data colored by the inferred loadings of the hierarchy. They are very similar to the true generative loadings.

In Fig. 5, we show the correlations between the learned loadings and the ground truth loadings for each of the 4 tested methods on the 7-dimensional data. BTF is able to recover the correct factor loadings and most of the hierarchical structure, although the position of two factors in the hierarchy are switched, demonstrating the identifiability challenge in recovering ground truth. However, the differences between BTF and PCA are stark; PCA recovers an entirely different set of factors, and the first two sets of loadings account for the majority of the correlation to all of the ground truth loadings. The hierarchical structure is not recovered and the factors do not reflect any individual ground truth structural elements. ICA performs only slightly better, presumably due to the relaxation of the orthogonality constraint. In our experiments, we found that TCA correctly identified that

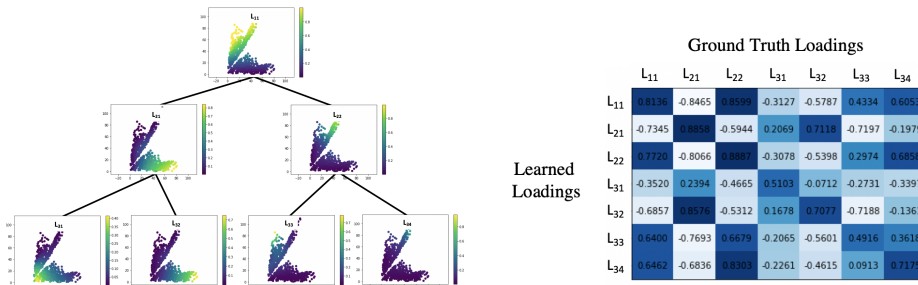

Figure 4: *Left:* A visualization of the effects learned by BTF when applied to the 2-dimensional simulation, arranged in the hierarchy induced by the generative loading structure. Each visualization shows the data colored by the corresponding learned loading in the hierarchy. *Right:* A heatmap showing the Spearman correlation statistics of the ground truth loadings to the BTF loadings when applied to the 2-dimensional simulation.

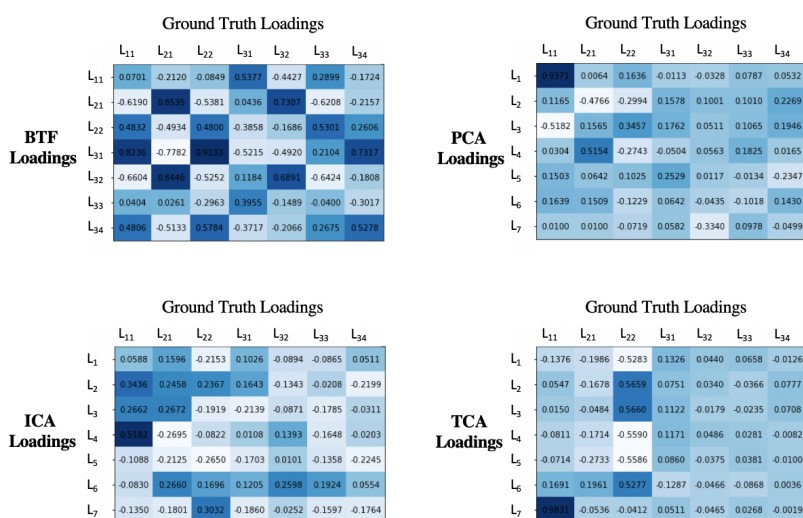

Figure 5: The correlations between learned loadings and ground truth loadings for each of the 4 tested methods on the 7-dimensional simulation data.

there are three levels to the hierarchy. However, it is unable to identify the true binary structure of the hierarchy and the loadings it recovers do not correlate well with ground truth.

None of the baseline approaches provide an interpretation of the functional relationships between the structural elements beyond parent-child structure (in the case of TCA); no conditional dependencies are explicitly represented by the loadings as they are in BTF. This demonstrates one of the unique advantages of BTF.

## 4 APPLICATIONS TO GENE EXPRESSION AND CLINICAL DATA IN BREAST CANCER PATIENTS

In order to test BTF in a real-world application, we applied single-view (SV) and multi-view (MV) BTF to a clinical and genomic dataset (Curtis et al., 2012) containing gene expression and clinical measurements collected from 2,000 breast cancer patients. We hypothesized that leveraging the multi-modal aspect of the data using multi-view BTF could help to address the aforementioned concerns about identifiability.

To even further increase the identifiability of the model and to aid in our analysis, we adjusted the structure of the loading compositions in the MV-BTF model such that the top level of the hierarchy includes 4 sub-trees. The means of the Gaussian distributions generating the data were adjusted to linear combinations of 4 expressions, each one of the form specified by BTF. The loadings on each of the 4 sub-tree expressions were fixed to binary indicators of the prognostic PAM50 (Bernard et al., 2009) molecular subtype membership, thus ensuring that each subtree only captures signal in its corresponding subtype. The first 3 levels of the resulting hierarchy is shown in Fig. 4.

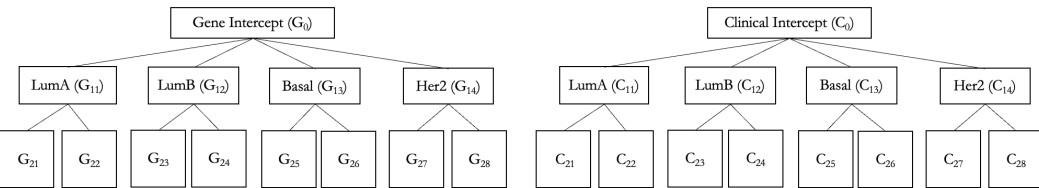

Figure 6: The first 3 levels of the modified subtype-specific structure inferred in the METABRIC data.

Notably, BTF can optionally be run with non-negativity constraints. Because we wanted a high-fidelity model of gene expression, which is measured on an intrinsically positive scale, we applied the non-negative formulation of BTF to this data; thus, all loading and factor values were positively constrained during optimization. We describe our data pre-processing procedure in detail in the Supplementary Methods.

## 4.1 Evaluation Approach

To evaluate our single and multi-view results, we used a combination of visualization and gene set enrichment analysis. The clinical factors are easily visualizable and interpretable due to the low dimensionality of the data. The scaled clinical factors of our multi-view analysis are shown in Supplementary Fig. 12. In order to interpret the gene factors we used PreRanked Gene Set Enrichment Analysis (Subramanian et al., 2005), which evaluates a predefined ranking to determine whether different sets of genes are over-represented at either end of the scale defined by the ordered ranking when compared to a randomly permuted baseline. We treated the factors as rankings of the genes. Because lower factor values contribute less signal in a non-negative factorization of the data, we only considered significant enrichments where the Normalized Enrichment Score (NES) was greater than 0, limiting our analysis to gene sets that are over-represented in the higher factor values. Enrichments were evaluated for significance using 1,000 permutations and were corrected for multiple hypothesis testing using the Benjamini Hochberg approach at $\alpha = 0.1$.

To evaluate the performance of SV-BTF, we compared the number of unique biological enrichments of MSigDB canonical pathways (Liberzon et al., 2011) captured by the learned BTF factors to those captured by three baseline methods (PCA, ICA and NMF). We also evaluated the reconstruction errors of the 4 methods.

We evaluated the improvement of MV-BTF over SV-BTF by comparing the number of MSigDB oncogenic geneset enrichments (Liberzon et al., 2011) learned in each application. We also did a qualitative examination of the enrichments discovered by the multi-view approach as well as how they relate to patient trends in the learned hierarchical structure of the clinical view.

To better understand the degree to which transfer learning occurs in MV-BTF, we ran additional experiments wherein we added varying levels of Gaussian noise to the data. To this end, we learned 8 individual multi-view and single-view models with added Gaussian noises of $\sigma = 0.1, 0.2, 0.5$ and 1 added in each of the two views. All noised data was rescaled to ensure a fair comparison between views. In all analyses, models were fit 10 times with distinct random seeds so as to account for variability due to initialization. Additional information on our choice of model hyperparameters can be found in the Supplement.

## 4.2 RESULTS OF SINGLE-VIEW ANALYSIS

Fig. 7 shows the results of the application of BTF to the METABRIC expression data when compared to the 3 baseline approaches. The reconstruction errors of BTF and NMF in contrast to the other methods are somewhat higher. This suggests that the increase is due to the stronger assumptions made by the models, especially the similar non-negativity constraints.

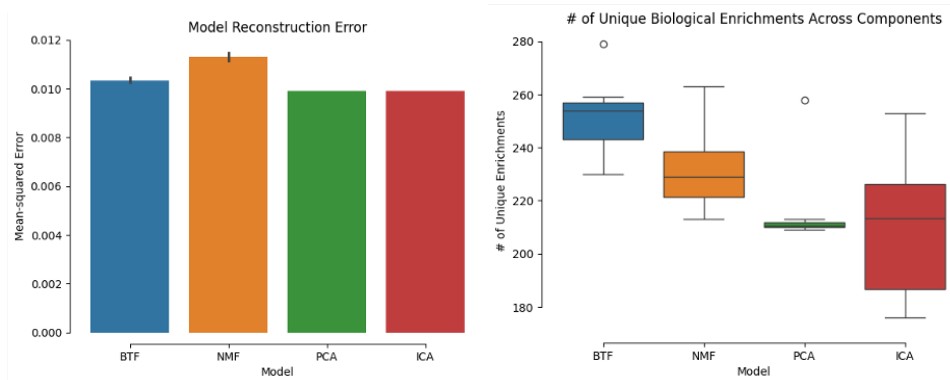

Figure 7: *Left:* The reconstruction error of BTF and 3 baseline models when applied to the METABRIC gene expression data. *Right:* The number of unique biological enrichments recovered in each of the 4 approaches when evaluating the factors using GSEA PreRank.

BTF outperforms the baseline approaches in number of high-ranking biological enrichments recovered. These results suggest that by incorporating a hierarchical representation of the dependencies inherent to the biological data, we are better able to recover the underlying biological signal. We hypothesize that the BTF approach could be particularly powerful in this context due to its explicit representation of opposing effects within the loading structure, which more closely resembles the natural up- or down-regulation of different genes in concert with specific biological mechanisms.

When we relax the restriction of our biological enrichments and consider genesets that are significantly overrepresented at the bottom of the rankings defined by the factor means, we note that the non-negative approaches (NMF and BTF) recover fewer enrichments than the other approaches (PCA and ICA) (See supplementary Fig. 10); while a non-negative model is likely to be a higher-fidelity representation of the true underlying biology, this result highlights the challenge of recovering biological enrichment under strong constraints.

## 4.3 RESULTS OF MULTI-VIEW ANALYSIS

Next, we evaluated the performance of multi-view BTF with respect to cancer-specific geneset enrichments. The enrichment results for each experiment are shown in Fig. 8. We found that the multi-view approach captured more cancer-specific enrichments than the single-view approach. This demonstrates that the model is effectively able to leverage a second view in order to improve its representation of the structure in the first. Additionally, we found that MV-BTF captures more enrichments than SV-BTF even when we add small amounts of artificial noise to the expression view. In these cases, fewer enrichments are captured by SV-BTF than in the noiseless data, but MV-BTF increases the enrichment signal nonetheless. When more noise is added, the clinical signal eventually dominates the learning process, resulting in fewer enrichments. We also see found that BTF is able to leverage even noisy clinical data to recover more enrichments in the biological data. This suggests that multi-view BTF is a potentially powerful tool for the analysis of confounded or noisy data, as is often the case in biological applications, although care needs to be taken to weight the views appropriately. We found that these trends persisted with respect to a broader set of biological enrichments (see Supplementary Fig. 11), although variation between noise-levels was less predictable.

The enrichments in each subtype sub-tree demonstrate that each tree captures informative and highly-interpretable enrichments that reflect real relationships between biological mechanisms and the clinical trends in that subtype. In the interest of space, we analyze and visualize this capability in

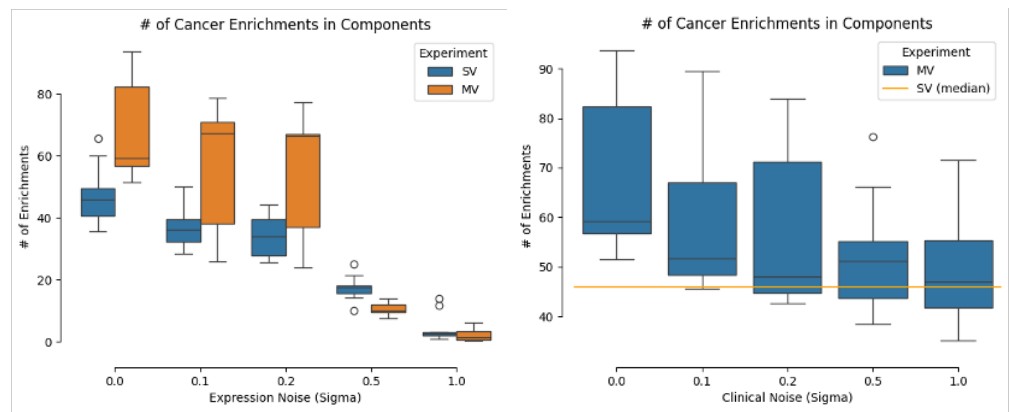

Figure 8: *Left:* The numbers of PreRank GSEA cancer-specific geneset enrichments represented in the factors of both the SV and the MV approaches, for varying levels of added noise in the expression view. *Right:* Cancer-specific geneset enrichments for varying added noise in the clinical view.

the luminal A subtype (Fig. 9) subtree. The root nodes of the trees function like intercepts, and thus describe the global effects that define the molecular subtype. True to the directional loading structure of BTF, the right child describes biological effects common to the progression of most luminal A breast cancers, such as estrogen signaling and estrogen dependent gene expression, which play a major role in luminal breast cancers (Clusan et al., 2023) and insulin secretion, which is known to promote estrogen-dependent breast cancers (Rose & Vona-Davis, 2012). It is also enriched for pathways such as FOXA1 signaling, a crucial factor in the progression of breast cancer that is also associated with better prognosis in the luminal A subtype. (Metovic et al., 2022) In addition to estrogen-receptor (ER) positivity, post-menopausal status and age, mucinous tumors are also significantly up-weighted in the corresponding clinical factor; while relatively rare, they occur exclusively in luminal A breast cancers (Limaiem & Ahmad, 2023). In contrast, the left child enrichments relate to biological mechanisms of poor prognosis due to treatment resistance; these cases are exceptional given that this subtype is the least aggressive molecular subtype and generally responds well to treatment (Orrantia-Borunda et al., 2022). Consistent with this interpretation, the corresponding clinical factor suggests older patients with larger tumors and higher rates of morbidity.

The children of the left branch are enriched for biological processes that are responsible for tumor invasion and metastasis, including extra-cellular matrix organization (Elgundi et al., 2020) and collagen formation (Zhang et al., 2023), as well as different known mechanisms of treatment resistance: for example, lipid metabolism (left), which is implicated in driving resistance to endocrine therapy in invasive lobular carcinoma (ILC) (Du et al., 2018), versus Wnt pathway signaling (right) (Xu et al., 2020). Notably, the corresponding clinical factors accurately capture related clinical effects and their directionality: high cellularity, low survival patients on the right, and ILC patients on the left. Recent work suggests that patient survival with ILC is highly time-dependent and also depends on menopausal status, age and ER status. (Chamalidou et al., 2021)

The children of the right clinical branch differentiate a continued trajectory of tumor growth (with higher grade tumors and higher Nottingham Prognostic Index (Haybittle et al., 1982) scores) from progesterone-receptor (PR) positive cases, which generally have better prognosis (Prat et al., 2013). The corresponding biological factor is not enriched for any genesets; however, Hashmi et al. (2018) suggest that more than 80% of luminal A breast cancers may be PR+, and thus we hypothesize that much of the relevant biological signal may be captured by its parent factor. The other child biological factor is enriched for many biological mechanisms related to tumor growth, including growth factor activation (Witsch et al., 2010) and cell cycle machinery (Thu et al., 2018), as well as specific pathways that are recent and currently active drug targets including C-MYC transcriptional activation (Llombart & Mansour, 2022) and Aurora B kinase signaling. (Borah & Reddy, 2021)

Overall, these results demonstrate that MV-BTF can be used to understand, investigate and interpret the real relationships between patient biology and clinical presentation. Additionally, MV-BTF could be a valuable tool for hypothesis generation; for example, to propose specific biological mech-

anisms that could underlie patient trends or outcomes. We include the exact biological enrichments and corresponding clinical factors for all 4 subtrees in the Supplementary material.

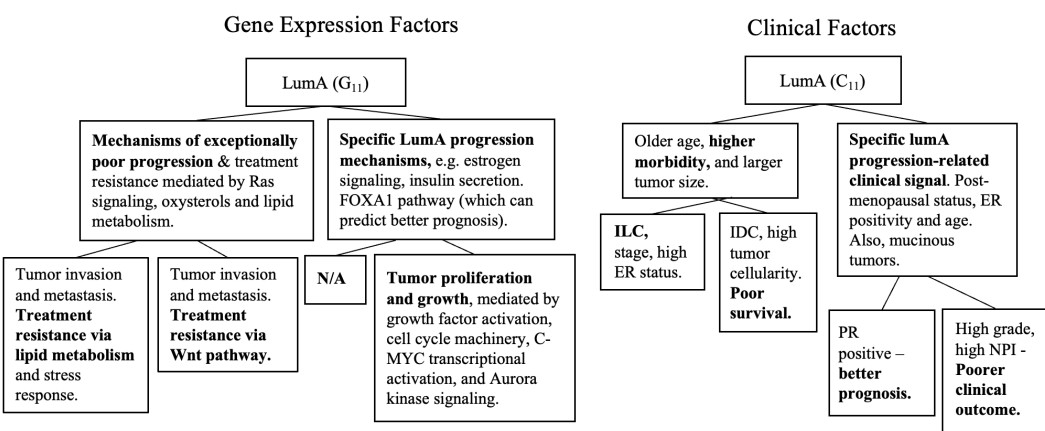

Figure 9: Biological mechanisms and corresponding clinical effects represented by the learned MV-BTF structure in the luminal A subtype.

## 5 DISCUSSION

Commonly used factorization approaches fail to account for the complexities of many real world datasets containing both local and global effects with higher order dependencies between factors. We show that Bayesian Tree-dependent Factorization is a Bayesian approach capable of interpretably and effectively discovering dependent, hierarchical structure that captures both specific and broader effects. One advantage of BTF is that it is highly flexible and allows for the specification of various priors on the latent factors. An additional strength of BTF is the continuous but constrained nature of the factor loadings, which allow the user to interpret the learned factors as continuous effects with loadings that indicate their conditional weight given their parents. We also demonstrate that we can construct a multi-view formulation of BTF that allows for integrative discovery of common mechanisms across modalities. Furthermore, our applications of BTF to breast cancer patient data demonstrate that BTF and MV-BTF are capable of uncovering and aiding in the interpretation of the real hierarchical mechanisms underlying complex datasets and thus could be valuable tools for investigation and hypothesis generation in contexts such as drug target discovery.

BTF is not without limitations. In particular, the model makes very strong assumptions about the nature of the relationships between dependent factors and how the hierarchy of these factors is structured. Due to the number of possible configurations of the underlying latent hierarchies, consideration of all the potential structures quickly becomes intractable. The binary tree learned by BTF is unlikely to always reflect the true underlying structure of the data, and care must be taken not to interpret the factor relationships too literally. Additionally, BTF requires that the user specify the depth of the learned factor tree *a priori*. Future work may attempt to relax these assumptions by automatically inferring properties of the tree structure. Similarly, our multi-view formulation of BTF makes the very strong assumption that the loadings across views be identical; future work could relax this assumption, for example by implementing a weighted likelihood that prefers factorizations with loadings that are highly correlated or otherwise related across views.

Broadly speaking, there are many opportunities for advancement in hierarchical factorization methods. In particular, many methods (including BTF) rely on prescriptive assumptions regarding the form of the higher order dependencies between factors. Future work could attempt to incorporate a deep learning approach or other highly non-linear functional forms in order to relax these assumptions as much as possible and accommodate additional complexity. Overall, we expect that hierarchical approaches like BTF that prioritize the interpretability of higher-order dependencies and latent mechanistic structure will continue to serve an important function in real world applications, particularly in critical decision making tasks.

REPRODUCIBILITY

Complete supporting this publication (including model and optimization code, simulation generation, and visualizations) will be made available via Github at the time of publication. For now, we include partial code specifying the model in the Supplementary material, so as to aid the review process.

This study makes use of data generated by the Molecular Taxonomy of Breast Cancer International Consortium. Funding for the project was provided by Cancer Research UK and the British Columbia Cancer Agency Branch.

The METABRIC datasets are accessible upon request from the European Genome-Phenome Archive (EGA) using the accession number EGAS00000000083. Corresponding clinical data is available from the corresponding publication. (Curtis et al., 2012)

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

## 6 SUPPLEMENT

### 6.1 PREPROCESSING OF THE METABRIC DATA

Before applying BTF, we pre-processed the data. We log-transformed the quantile-normalized expression data and then generated approximate log2 gene expression intensities from the means of all probe intensities occuring within the gene bodies. We filtered to the top 10% most variably expressing genes over samples (2,011 genes). We also made sure to include all available genes from the PAM50 set (Bernard et al., 2009), which are commonly used as molecular markers for breast cancer subtype. We imputed 22 missing values using a nearest neighbors approach with five neighbors.

In the clinical data, we converted categorical values to one-hot encoding and then chose a subset of the features based on relevance and low degrees of missingness. Among the features included were histological subtype, PAM50 subtype, receptor status, stage, grade, tumor size, cellularity, age at diagnosis and tumor size. We also included survival status and the right-censored time to survival in days. The full set of features used in this analysis is shown in Fig. 12.

We removed patient samples of the Normal-like PAM50 subtype to reduce the odds of confounding the other breast cancer subtypes; the true nature of this subtype is not well understood and it was originally defined using normal breast tissue samples for the purpose of quality control. (Bernard et al., 2009)

Clinical data were scaled using a min-max scaler to the range [0, 1] so as to ensure positive values for the purpose of the multi-view analysis.

### 6.2 MODEL HYPERPARAMETERS

We ran all formulations of BTF with the following hyperparameter settings:

$$\theta = 1$$
$$\beta_0 = (10, 10)$$
$$\sigma_0 = 0.01$$

Multi-view models were run with model weights of 1 and 15 on the expression and clinical views respectively; we chose these weights because we found that they effectively showcased the trade-off between performance improvement and the effect of the added noise in our multi-view experiments.

Step sizes used for the ADAM optimization algorithm were: 0.1 (loadings), 0.1 (factors), and 0.01 (global noise parameter).

Factors were initialized to random samples. Loadings were initialized to uniform vectors of 0.5 (an uninformed choice consistent with our interpretation of the loadings as conditional weights between 0 and 1). The global noise parameter was initialized to 0.1.

Model optimization ran for 15,000 iterations in each experiment to ensure convergence.

## 6.3 SUPPLEMENTARY FIGURES

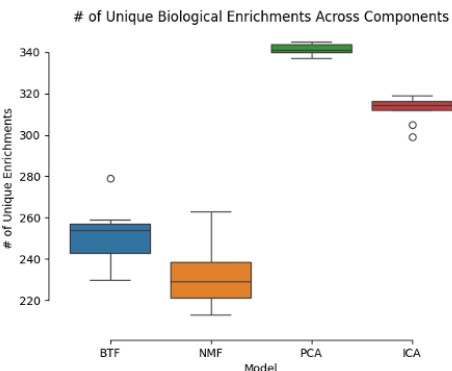

Figure 10: *Left:* The reconstruction error of BTF and 3 baseline models when applied to the METABRIC gene expression data. *Right:* The number of unique biological enrichments recovered in each of the 4 approaches when evaluating the factors using GSEA PreRank.

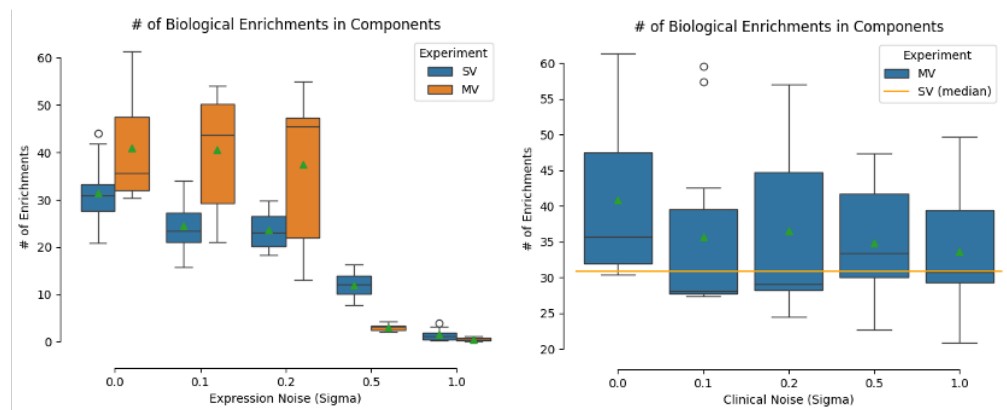

Figure 11: *Left:* The numbers of PreRank GSEA biological enrichments represented in the factors of both the single-view and the multi-view approaches, compared to models learned with varying levels of added noise in the expression view. *Right:* Biological enrichments for varying levels of added noise in the clinical view. Green arrows depict means.

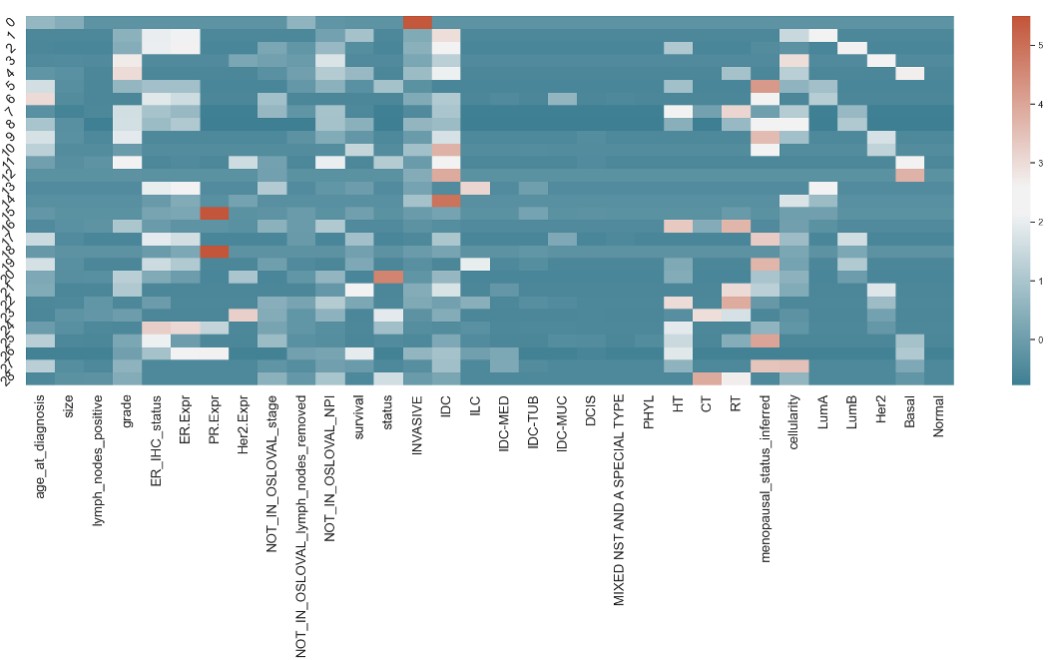

Figure 12: A heatmap showing learned factors in the clinical view of the MV-BTF run used for the luminal A subtree analysis. Factors are scaled for easier interpretation. The factor values of the molecular subtype features (LumA, LumB, Her2, and Basal) reflect the subtree structure (e.g. factors 1, 2, 3, and 4 are the root nodes of each subtree, and factors 5 and 6 are the top-level child nodes of the luminal A subtree).

