# OpenReview forum: "Bayesian Tree-Dependent Factorization"
_ICLR.cc/2025/Conference — ICLR 2025 Conference Withdrawn Submission_

### Official Review · Reviewer_aEma · 2024-10-15

**Soundness:** 1
**Presentation:** 2
**Contribution:** 1
**Rating:** 3
**Confidence:** 4

**Summary:**

The authors' work tackles the problem of learning factor models in which the factors/loadings have tree structure. The problem itself is interesting. However, this work is not properly finished and misses many details that should be present.
While I acknowledge that the length of the manuscript is somehow limited, these details could be added in the supplement. I think this work is interesting, but it needs substantial polishing before it is presented.

**Strengths:**

Interesting topic and model.

**Weaknesses:**

A simple example is the ADAM citation on page 4, which is not well formatted. It should be Kingma and Ba 2014, not Diederik 2014.

The authors'  mention  the multi-view version of their method multiple time. However, I can't find the model description anywhere. Please for the next iteration of the manuscript just refer to the supplementary material and describe this model.

Similarly, the authors wrote:
"Notably, BTF can optionally be run with non-negativity constraints. Because we wanted a high-
fidelity model of gene expression, which is measured on an intrinsically positive scale, we applied
the non-negative formulation of BTF to this data; thus, all loading and factor values were positively
constrained during optimization."

Is this another model as the one described in the main manuscript? If yes, this model should be described (do you use non-negative inducing prior), or do you just hard threshold the posterior mean?



I am quite unsure about the fit of this model using ADAM. While this kind of off-the-self optimizer can be good, it can work really poorly as well (e.g., the EM is much better than ADAM to fit mixture); in such a setting, there are other proper tools that lead to much better estimates that are not computationally demanding (see 10.1080/01621459.2017.1286241 for single factor fitting procedure in a similar type of tree structure, I am not one of the authors). I am a bit nervous about fitting this model using ADAM, as there is a substantial literature approximate inference on such a model (see this Empirical Bayes Matrix Factorization JMLR 2021 for a simpler version), and it seems too good to be true that such complex model could properly be estimated using ADAM. My understanding is that using ADAM one can get a MAP estimate of this model or get an approximate solution but not the entire posterior distribution. The authors are not clear about what they actually estimate; the loss function should be written down, or the Variational approximation they are optimizing over.

Again, while I think the topic is interesting  and the simulation and application might be correct, the current state of the paper is not suitable for publication.

**Questions:**

See my weakneses report

---

### Official Review · Reviewer_obhP · 2024-10-26

**Soundness:** 2
**Presentation:** 2
**Contribution:** 2
**Rating:** 3
**Confidence:** 3

**Summary:**

The goal of this paper is to develop the method of Bayesian tree-dependent factorization (BTF) and its multi-view extension. BTF aims to construct a tree-based model which can factor the data. Each factor has a conditional relationship with its parent. As a result, it has the potential to capture both global and local effects. By developing the multi-view extension for BTF, the authors perform joint analysis of multiple data modalities to provide some insights on relationships among different data types.

**Strengths:**

Well-known methods like principal component analysis and independent component analysis provide orthogonal factors, but do not offer hierarchical structure for the dependencies among effects. This paper proposes a Bayesian approach, Bayesian tree-dependent factorization (BTF), for unified factor analysis of hierarchical structured continuous effects.

**Weaknesses:**

Although the goal is ambitious, the paper unfortunately didn’t provide a convincing method and explanation. The short explanation of the model reflects cumbersome notations and doesn’t provide clear interpretation as it was argued. Furthermore, the key contribution of the multi-view extension only has a couple of short paragraphs without many details. So, it is hard to appreciate the contribution and challenges involved for the problem. The authors may consider to add more explanations on the interpretations of the factors and loadings on an illustrative example. For the multi-view example, give more details on what's the challenges involved due to multiple modalities of data and key steps to address the challenges.

 The important computational aspect was just mentioned with one or two sentences. The authors may consider to give a sketch of the algorithm or pseudo-code which shows the steps involved in the computation.

Furthermore, the writing of the paper needs to be significantly improved. For example, many references are placed incorrectly (should be before the period instead of after etc.). To list a few, on the bottom of page 1, "fully continuous effects. (Sugahara & Okamoto, 2024;
Almutairi et al., 2021; Li et al., 2019)"; On the second paragraph of Page 2, "...rely on ensembles of decision trees
(or random forests). (Breiman, 2001)"

**Questions:**

1.	The authors argue that BTF can be more interpretable than methods like PCA, ICA. PCA and ICA can typically handle quite large dimensional problems for biomedical applications. How large dimensions can BTF handle? What about the computational cost comparisons? The authors may provide empirical runtime comparisons on datasets of increasing dimensionality, and discuss the computational complexity of BTF compared to PCA/ICA.
2.	The simulation comparison is mainly for visualizing the loadings among different methods. It can be difficult to judge by just looking at different loading matrices. It can be helpful to use some summary metrics to compare them. For example, using certain matrix norms for the differences between the estimated and true loading matrices. In addition, it can be helpful to consider higher dimensional problems as well. Additional simulations to illustrate performance with higher dimensional problems are needed to be more aligned with real data applications. Please add computational time comparisons as well.
3.	The authors argue that BTF offers hierarchical interpretations in contrast to methods like PCA and ICA. However, how can one know whether such structure is valid for a given problem? What if the assumption is wrong? The authors should give detailed explanations on these. Some simulations on mis-specified settings are needed. For example, when there is no hierarchical structure or different structure than specified, it will be useful to show the sensitivity of the BTF method to such mis-specifications.
4.	There are many existing papers on multi-view data which are not mentioned in the paper. For example, the following paper proposed a PCA extension method called JIVE for integrative analysis. The authors should discuss and compare such relevant papers in the literature thoroughly to demonstrate the clear advantage and disadvantages of the proposed method over existing methods in the literature. Perhaps a comparison table highlighting the key differences between BTF and methods like PCA, ICA, JIVE etc. would be useful.
https://arxiv.org/abs/1102.4110

---

### Official Review · Reviewer_exed · 2024-10-29

**Soundness:** 1
**Presentation:** 1
**Contribution:** 1
**Rating:** 1
**Confidence:** 3

**Summary:**

This paper puts forward a hierarchical factorization model. The model posits a complete binary tree, where each node of the binary tree has a d-dimensional factor, and each edge of the loading has a [0,1]-valued weight. Then the conditional likelihood of the data is a multivariate normal distribution such that the mean of the distribution is a linear combination of the factors, and the weight of each factor is the product of weights along the edges of the path from the root of the tree to the corresponding node. The covariance is not specified. The paper also puts forward a multi-view version of the model in which the number of factors grows with the number of views, but the loadings are fixed as equal.

The paper goes on to evaluate the method on simulated and real data.

**Strengths:**

On the positive side, the paper does a thorough examination of their model results on the gene expression and clinical data application.

**Weaknesses:**

This paper does a poor job at (a) motivating the proposed model, (b) specifying their model, (c) tying their work into the existing factor analysis literature, and (d) evaluating their methodology.

On (a), the paper does not discuss what the desired structure of the factors and loadings should be. The abstract states that the method uncovers hierarchical factors, but the paper does not specify what "hierarchical factors" look like. The stick-breaking structure of the loadings does not seem by itself to enforce any type of hierarchical structure on the loadings, as the model is unidentifiable.

As an example, one could consider the case z_11 = 1/2, z_21 = 1/2, z_22 = 1/2, F_11 = (0,0), F_12 = (1,0), F_22 = (0,1). This might be seen as having hierarchical structure on the factors. But the parameters z_11 = 1/8, z_21 = 2/7, z_22 = 1, F_11 = (0,1), F_12 = (1,0), F_22 = (0,1) have the same exact likelihood, and I would not think of the factors of being hierarchically organized.

The authors need to explain (1) the type of hierarchical organization they expect to see in the factors and (2) how their proposed leads to that structure.

On (b), I cannot tell what the prior distribution on z_ij is. Perhaps it is a Beta distribution, but the Beta distribution is generally specified by 2 parameters as opposed to 1. More importantly, the paper does not explain how their model is fit, only that ADAM is employed. Does this mean that the model is fit using variational inference? If so, what variational family is used? Or perhaps the model is fit using MAP estimation. But if that is the case, why?

On (c), the paper does not tie itself into the rich existing literature on factor analysis, in particular in non-parametric factor analysis. Two influential Bayesian approaches here are the Indian Buffet Process [Griffiths and Ghahramani, 2005] and the Multiplicative Gamma Process [Bhattacharya and Dunson, 2011]. In fact, there has even been a tree-structured factor model proposed that encodes a hierarchical relationship amongst the factors [Zhang et al., 2011]. This literature should be discussed in the paper.


[Griffiths and Ghahramani, 2005] Infinite Latent Feature Models and the Indian Buffet Process. Neurips, 2005.

[Bhattacharya and Dunson, 2011] Sparse Bayesian infinite factor models. Biometrika, 2011.

[Zhang et al., 2011] Tree-Structured Infinite Sparse Factor Model. ICML, 2011.


On (d), I'm not at all convinced by the simulations. The simulation study generates a single dataset of 1K points in 7-dimensions and plots Spearman cross-correlation confusion matrices. I don't know how to interpret this. The paper needs to come up with quantitative metrics to compare their baselines on, scale along different problem settings (dimension, data size, aspects of generative process), and show how each of the models perform in those settings.

**Questions:**

How is the model fit? VI? MAP inference?

Mathematically, how does the structure of loadings lead to hierarchically-organized factors?

---

### Official Review · Reviewer_gsSk · 2024-11-02

**Soundness:** 2
**Presentation:** 2
**Contribution:** 2
**Rating:** 3
**Confidence:** 4

**Summary:**

A Bayesian factorization method is presented for finding hierarchical
factor structure, such that child nodes in a tree are modelling
sub-structure of the parent. The method is extended to multiple data
sources, and applied to molecular biological data.

**Strengths:**

The method is a sensible though restricted generalization of earlier
methods.

A very positive aspect is that it has been tested on other case
studies than only usual-suspect benchmark cases.

The limiations have been discussed very openly in the Discussion,
including the very restrictive assumptions in the methods.

**Weaknesses:**

The method has only been compared to very old factorization
methods.

The method makes very restrictive assumptions: in the multi-view
extension the loadings are equal in the different views. The imposed
tree structure for the factors is very rigid and needs to be specified
by the user.

Presentation of the simulation results could be clearer; it is not
obvious what we should infer from Figs 4 and 5.

Not clear what the components are good for, precisely.

Minor comment: Notation has not been introduced before being used in
Fig 1.

**Questions:**

1. Where does the structural knowledge needed for Fig 6 come from, and
how generally available is such knowledge in other applications?

2. What will the factors be used for, and how do we judge if they are
good for that purpose?

3. Is there a real use case, where the extracted components are actually
needed? And if yes, could task performance be measured to quantify
relative performance of the methods?

4. Why are the biological enrichments a good measure of goodness?

5. Are there really no newer methods to compare against?

6. What does this sentence in discussion mean: "Care must be taken not to
interpret the factor relationships too literally"? How should the
factors be interpreted then?

---

### Author Response · Authors · 2024-12-02
**Thank you for your reviews**

We appreciate the time that the reviewers took to provide us feedback on this work. Based on this feedback, we are withdrawing this submission, and we will consider your points as we update the manuscript for another venue.

---

### Note · Authors · 2024-12-02

I have read and agree with the venue's withdrawal policy on behalf of myself and my co-authors.